# Modulation of *Prdm9*-controlled meiotic chromosome asynapsis overrides hybrid sterility in mice

Sona Gregorova[1†], Vaclav Gergelits[1†], Irena Chvatalova[1‡],
Tanmoy Bhattacharyya[1§], Barbora Valiskova[1,2], Vladana Fotopulosova[1],
Petr Jansa[1], Diana Wiatrowska[1], Jiri Forejt[1*]

[1]Institute of Molecular Genetics, Academy of Sciences of the Czech Republic, Vestec, Czech Republic; [2]Faculty of Science, Charles University, Prague, Czech Republic

**\*For correspondence:**
jforejt@img.cas.cz

[†]These authors contributed equally to this work

Present address: [‡]The National Institute of Public Health, Prague, Czech Republic; [§]The Jackson Laboratory, Bar Harbor, United States

**Competing interests:** The authors declare that no competing interests exist.

**Abstract** Hybrid sterility is one of the reproductive isolation mechanisms leading to speciation. *Prdm9*, the only known vertebrate hybrid-sterility gene, causes failure of meiotic chromosome synapsis and infertility in male hybrids that are the offspring of two mouse subspecies. Within species, *Prdm9* determines the sites of programmed DNA double-strand breaks (DSBs) and meiotic recombination hotspots. To investigate the relation between *Prdm9*-controlled meiotic arrest and asynapsis, we inserted random stretches of consubspecific homology on several autosomal pairs in sterile hybrids, and analyzed their ability to form synaptonemal complexes and to rescue male fertility. Twenty-seven or more megabases of consubspecific (belonging to the same subspecies) homology fully restored synapsis in a given autosomal pair, and we predicted that two or more DSBs within symmetric hotspots per chromosome are necessary for successful meiosis. We hypothesize that impaired recombination between evolutionarily diverged chromosomes could function as one of the mechanisms of hybrid sterility occurring in various sexually reproducing species.

DOI: https://doi.org/10.7554/eLife.34282.001

## Introduction

Hybrid sterility (HS) is a postzygotic reproductive isolation mechanism that enforces speciation by restricting gene flow between related taxa. HS is a universal phenomenon observed in many eukaryotic inter-species hybrids, including examples in yeast, plants, insects, birds, and mammals (*Coyne and Orr, 2004*; *Maheshwari and Barbash, 2011*). In the early days of genetics, HS was difficult to accommodate in Darwin's theory of evolution by natural selection. In time, however, the Bateson–Dobzhansky–Muller incompatibility (BDMI) hypothesis (*Muller and Pontecorvo, 1942*; *Dobzhansky, 1951*; *Orr, 1996*) explicated HS, and more generally any hybrid incompatibility, as a consequence of the independent divergence of mutually interacting genes resulting in aberrant interaction of the new alleles that have not been tested by natural selection. HS has several common features across various sexually reproducing eukaryotic species. Haldane's rule posits that if one sex of the $F_1$ offspring of two different animal races is absent, rare, or sterile, it is the heterogametic sex (XY or ZW) (*Haldane, 1922*). Another common feature refers to the disproportionately large role of Chr X compared to that of autosomes in reproductive isolation (*Presgraves, 2008*). More recently, interaction between selfish genomic elements causing meiotic drive and their suppressors has been implicated in some instances of reproductive isolation (*Orr, 2005*; *Zhang et al., 2015*).

The molecular mechanisms underlying HS remain an unresolved question. Historically, genic and chromosomal mechanisms of HS had been hypothesized, but the latter were soon dismissed as

**eLife digest** It has been known for centuries that hybrids between closely related species are often infertile. This hybrid sterility was an enigma for Charles Darwin, who understood that it influenced how new species formed but could not fit it with his theory of evolution by natural selection.

Sex cells – in mammals, the egg or sperm cells – form by a process called meiosis. During meiosis, chromosomes formed of DNA inherited from the mother pair up with the equivalent chromosomes inherited from the father and exchange sections of their DNA. This process is called synapsis and homologous recombination.

A gene called *Prdm9* determines where the DNA will break and recombine. *Prdm9* plays a major role in determining whether the male hybrid offspring of two laboratory strains of mice (which come from different subspecies) are sterile. In sterile hybrids, the two versions of *Prdm9* interact in ways that disturb the DNA repair process. However, these interactions are not enough on their own to cause hybrid males to be sterile. The currently prevailing view is that interactions between a large number of other – currently unidentified – genes also contribute to sterility. But could there be other processes involved that do not involve gene interactions?

To investigate, Gregorova, Gergelits et al. utilized strains of hybrid mice where a pair of chromosomes of one subspecies was substituted by the corresponding pair from the other subspecies. This generated hybrids with stretches of DNA that came entirely from a single subspecies. Having such a stretch that lasted for 27 million or more DNA base pairs fully restored synapsis in a given pair of chromosomes during meiosis. Hybrid sterility was reversed when synapsis was restored in the four chromosomes that were most strongly affected by synapsis not occurring.

The results presented by Gregorova, Gergelits et al. provide a direct link between *Prdm9*-controlled chromosome synapsis and the interruption of meiosis. Hybrid sterility occurs in all sexually reproducing organisms, as does chromosome pairing during meiosis. Thus *Prdm9* could control a particular case of a more universal mechanism that enables new species to form.

DOI: https://doi.org/10.7554/eLife.34282.002

unlikely on the grounds that large chromosomal rearrangements do not segregate with HS genetic factors (*Dobzhansky, 1951*). Other possible forms of non-genic chromosomal HS were not considered because of the limited knowledge of the carrier of genetic information at the time. Thus, for the past 80 years or so, the focus on the genic control of HS prevailed (*Dobzhansky, 1951*; *Orr, 1996*; *Forsdyke, 2017*). In studies mapping HS genes, the *Drosophila* group of species has been the model of choice, yet only five *Drosophila* HS genes, namely *OdsH*, *JYAlpha*, *Ovd*, *agt,* and *Taf1*, have been identified to date, none of which has a known interacting partner predicted by the BDMI hypothesis (*Ting et al., 1998*; *Masly et al., 2006*; *Phadnis and Orr, 2009*). The low success rate of the positional cloning of HS genes was explained by the oligogenic or polygenic nature of HS phenotypes and by the inherent difficulty in genetically dissecting the phenotype that prevents its own transfer to progeny.

Over 40 years ago, we introduced the house mouse (*Mus musculus*) as a mammalian model for the genetic analysis of HS. The first mouse HS locus *Hst1* was genetically mapped in crosses of laboratory inbred strains (predominantly of *Mus musculus domesticus* (*Mmd*) origin) with wild *Mus musculus musculus* (*Mmm*) mice (*Forejt and Iványi, 1974*). Later, we developed the PWD/Ph and PWK/Ph inbred strains purely from the wild *Mmm* mice of Central Bohemia (*Gregorová and Forejt, 2000*) and used them in the positional cloning of *Hst1* by high-resolution genetic crosses and physical mapping (*Gregorová et al., 1996*; *Trachtulec et al., 1997*). Finally, we identified the *Hst1* locus with the PR-domain-containing nine gene (*Prdm9*) (*Mihola et al., 2009*), which codes for histone H3 lysine 4/lysine 36 methyltransferase (*Powers et al., 2016*) the first and still the only HS gene known in vertebrates. Most of the tested laboratory inbred strains share either the $Prdm9^{Dom2}$ or the $Prdm9^{Dom3}$ allele (*Parvanov et al., 2010*; *Brunschwig et al., 2012*). The former allele was found in inbred strains producing sterile male hybrids when crossed with PWD females, whereas the $Prdm9^{Dom3}$ was observed in the strains that yielded quasi-fertile males in the same type of inter-subspecific crosses (*Forejt et al., 2012*).

The male sterility of (PWD x C57BL/6)F1 (henceforth PB6F1) hybrids depends on the interaction of the heterozygous allelic combination $Prdm9^{Msc}/Prdm9^{Dom2}$ with the PWD allelic form of the X-linked Hybrid sterility X Chromosome 2 locus, $Hstx2^{Msc}$ (*Dzur-Gejdosova et al., 2012*; *Bhattacharyya et al., 2014*). For the sake of clarity and to stress the origin of the alleles, we will use the names $Prdm9^{PWD}$, $Prdm9^{B6}$ and $Hstx2^{PWD}$ in the rest of this paper. Any other tested allelic combination of these two major HS genes yields fully fertile or subfertile male hybrids (*Dzur-Gejdosova et al., 2012*; *Flachs et al., 2012*). The proper allelic combination of $Prdm9$ and $Hstx2$ genes is necessary but not sufficient to govern HS completely because less than 10% instead of the expected 25% of (PWD x B6) x B6 male backcross progeny replicated the infertility of male PB6F1 hybrids (*Dzur-Gejdosova et al., 2012*). Initially, we explained this 'missing heritability' by assuming the genic interaction of three or more additional HS genes with a small effect that had escaped the genetic screen (*Dzur-Gejdosova et al., 2012*). However, an alternative, non-genic explanation emerged from the analysis of meiotic phenotypes of sterile hybrids. We observed multiple unsynapsed autosomal pairs decorated by phosphorylated histone γH2AX as a mark of persisting unrepaired DNA double-strand breaks (DSBs) in approximately 90% of primary spermatocytes of infertile PB6F$_1$ hybrids. The asynapsis was accompanied by disturbed transcriptional inactivation of sex chromosomes at the first meiotic prophase (*Bhattacharyya et al., 2013, 2014*). The failure of intersubspecific homologs to synapse was clearly dependent on interhomolog interactions, and we suggested that their fast-evolving nongenic DNA divergence could be the causal factor. Because meiotic asynapses of different origin are known to compromise the normal progression of the first meiotic division (*Forejt, 1984, 1996*; *Mahadevaiah et al., 2008*; *Burgoyne et al., 2009*), we proposed that $Prdm9$ and $Hstx2$-directed asynapsis per se could be the ultimate cause of the sterility of male hybrids. Recently, the role of PRDM9 zinc-finger-domain binding sites within noncoding genomic DNA has been demonstrated in PB6F$_1$ male HS. Replacement of the mouse sequence encoding the PRDM9 zinc-finger array with the orthologous human sequence reversed sterility in (PWD x B6-$Prdm9^{Hu}$)F$_1$ hybrid males (*Davies et al., 2016*). In PB6F$_1$ hybrids, roughly 70% of the $Prdm9$-directed DSBs hotspots identified by the DMC1 ChIP-seq method were enriched on the 'nonself' homologous chromosome, as the DSB hotspots determined by the B6 allele of $Prdm9$ were found predominantly on PWD chromosomes, and vice versa. Such hotspots were designated as asymmetric DSB hotspots. Chromosome-specific quantification of asymmetry correlated well with the asynapsis rate across five arbitrarily chosen chromosomes of PB6F$_1$ hybrids (*Davies et al., 2016*; *Smagulova et al., 2016*). Another, non-exclusive interpretation of DMC1 ChIP-seq data pointed to significant enrichment of PRDM9-independent hotspots in the PB6F$_1$ hybrid testis, which occurs in promoters and other regulatory motifs and which is characteristic of spermatogenic arrest in $Prdm9$ knockout males (*Smagulova et al., 2016*). Recently, one third of PRDM9-dependent DSBs were reported within sequences that have at least some repetitive character, indicating that inappropriately high DSB levels in transposons and other repetitive elements may contribute to the infertility seen in some mouse hybrids (*Yamada et al., 2017*).

In this work, we studied the relationship between meiotic chromosome asynapsis, intersubspecific heterozygosity and male HS in a series of PB6F$_1$ hybrids carrying recombinant chromosomes with *Mmm/Mmm* consubpecific (belonging to the same subspecies) PWD/PWD homozygous intervals on *Mmm/Mmd* intersubpecific (belonging to different subspecies) PWD/B6 heterozygous background. We report the restoration of synapsis of intersubspecific chromosome pairs in the presence of 27 Mb or more of consubpecific sequence, and the reversal of HS by targeted suppression of asynapsis in the four most asynapsis-sensitive chromosomes. Our findings point to the chromosomal basis of $Prdm9$-directed hybrid male infertility as a (nonexclusive) alternative to a widely accepted concept of hybrid sterility driven by multiple genic incompatibilities.

## Results

### Small chromosomes are more susceptible to asynapsis in sterile F$_1$ hybrids

First, we ascertained the frequency of meiotic asynapsis separately for each chromosome pair of PB6F1 hybrid males by combining the use of fluorescence in-situ hybridization (FISH) to decorate chromatin from individual chromosomes with immunostaining of synaptonemal complex protein 3

(SYCP3) (a major component of axial/lateral elements), to visualize synaptonemal complexes, and HORMA domain-containing protein-2 (HORMAD2) (*Wojtasz et al., 2012*), to identify the axial elements of unsynapsed chromosomes (*Figure 1A*). Altogether, 4168 pachynemas from 40 PB6F1 hybrid males were analyzed. All autosomes of hybrid males displayed a certain degree of asynapsis, classified as complete, partial, or intermingled (more than two tangled univalents within labeled chromatin cloud), with frequencies ranging from 2.6% (Chr 1) to 42.2% (Chr 19) (*Figure 2—source data 1*). A strong bias was evident towards higher asynapsis rates in the five smallest autosomes (p=$5.2\times10^{-14}$, comparison of Generalized Linear Mixed Models [GLMM], *Figure 2A*). Recently, SPO11 oligos released during the processing of DSBs were sequenced, mapped and quantified at chromosome-wide scale in male mice of the B6 laboratory inbred strain (*Lange et al., 2016*). This information, together with the estimated frequency of asymmetric DSB hotspots in PB6F1 hybrids (*Davies et al., 2016*; *Smagulova et al., 2016*), enabled us to calculate the possible correlation between the number of DSBs within symmetric hotspots (hereafter symmetric DSBs) per chromosome per cell and synapsis between intersubspecific homologs. The calculation is based on and limited by the following premises: (i) the overall densities of DSBs on individual chromosomes of B6 and PB6F1 hybrid males are similar; (ii) approximately 250 DSBs occur per leptotene/zygotene cell (*Kauppi et al., 2013*); and (iii) the 0.28 proportion of symmetric DSB hotspots in (PWD x B6)F$_1$ hybrid males (*Davies et al., 2016*) is constant in all autosomes. Under these conditions, a strong negative correlation (Spearman's ρ=−0.760, p=0.0003) of asynapsis rate with predicted symmetric DSBs (*Lange et al., 2016*) can be seen (*Figure 2—source data 2*). This correlation is stronger than the correlation of the asynapsis rate with the chromosomal physical length (Spearman's ρ=−0.681, p=0.0013). Even though the chromosomal length and the expected number of symmetric DSB hotspots strongly correlate (Spearman's ρ=0.916, p=$1.1 \times 10^{-7}$), we observed that it is the symmetric DSB hotspots that affect the asynapsis rate. The chromosomal length does not add any additional explanation of the asynapsis rate to that provided by symmetric DSBs (p=0.709, comparison of GLMM models). On the contrary, the symmetric DSBs add an additional explanation of the asynapsis rate to that provided by the chromosomal length (p=0.046, comparison of GLMM models). Thus, our findings suggest that synapsis of a pair of homologous chromosomes depends on the presence of a certain minimum number of symmetric DSBs, as we elaborate further using a simulation described in the 'Discussion'.

Further, we examined the asynapsed chromosomes of PB6F$_1$ hybrids for localization of active chromatin using confocal fluorescence microscopy after Cot-1 RNA FISH (*Hall et al., 2014*) and HORMAD2 immunolabeling. Fluorescence signal quantification revealed that subnuclear regions of asynapsed chromosomes composed of sex chromosomes and/or autosomal univalents were lacking active euchromatin in contrast to other regions of the pachytene nuclei (*Figure 1—video 1*). We propose that the absence of active euchromatin is a consequence of the meiotic synapsis failure of intersubspecific chromosomes, known as meiotic silencing of unsynapsed chromatin (MSUC [*Burgoyne et al., 2009*]), which can act as an epigenetic component contributing to the meiotic phenotypes of sterile hybrids (*Larson et al., 2016*).

## The minimal length of consubspecific sequence necessary to rescue meiotic chromosome synapsis

We have shown previously that meiotic asynapsis affects intersubspecific (PWD/B6) but not consubspecific (PWD/PWD) pairs of homologous chromosomes in sterile male hybrids from crosses of PWD females and B6.PWD-Chr # consomic males (*Gregorová et al., 2008*; *Bhattacharyya et al., 2013, 2014*). Here, we searched for the minimum length of the PWD/PWD consubspecific sequence that still could secure synapsis of a chromosome and potentially restore fertility in the hybrids. Instead of replacing the whole B6 chromosome with its PWD homolog, we generated recombinant PWD/B6 and B6/PWD (centromere/telomere) chromosomes. To do that, we crossed the male hybrids between two B6.PWD-Chr # consomic strains with a PWD female and estimated the minimum size and location of consubspecific PWD/PWD stretches needed for synapsis rescue, as shown in *Figure 3A*. In three such generated 'two-chromosome crosses' (hereafter referred to as 2-chr crosses) we investigated the effect of the PWD/PWD consubspecific intervals on the asynapsis rate in six different chromosomes — two in a given experiment, namely Chr 5 and Chr 12 (*Figure 3—source datas 1* and *2*), Chr 7 and Chr 15 (*Figure 3—source datas 3* and *4*) and Chr 17 and Chr 18 (*Figure 3—source datas 5* and *6*). Altogether, 122 chromosomes from over 12,000 pachynemas

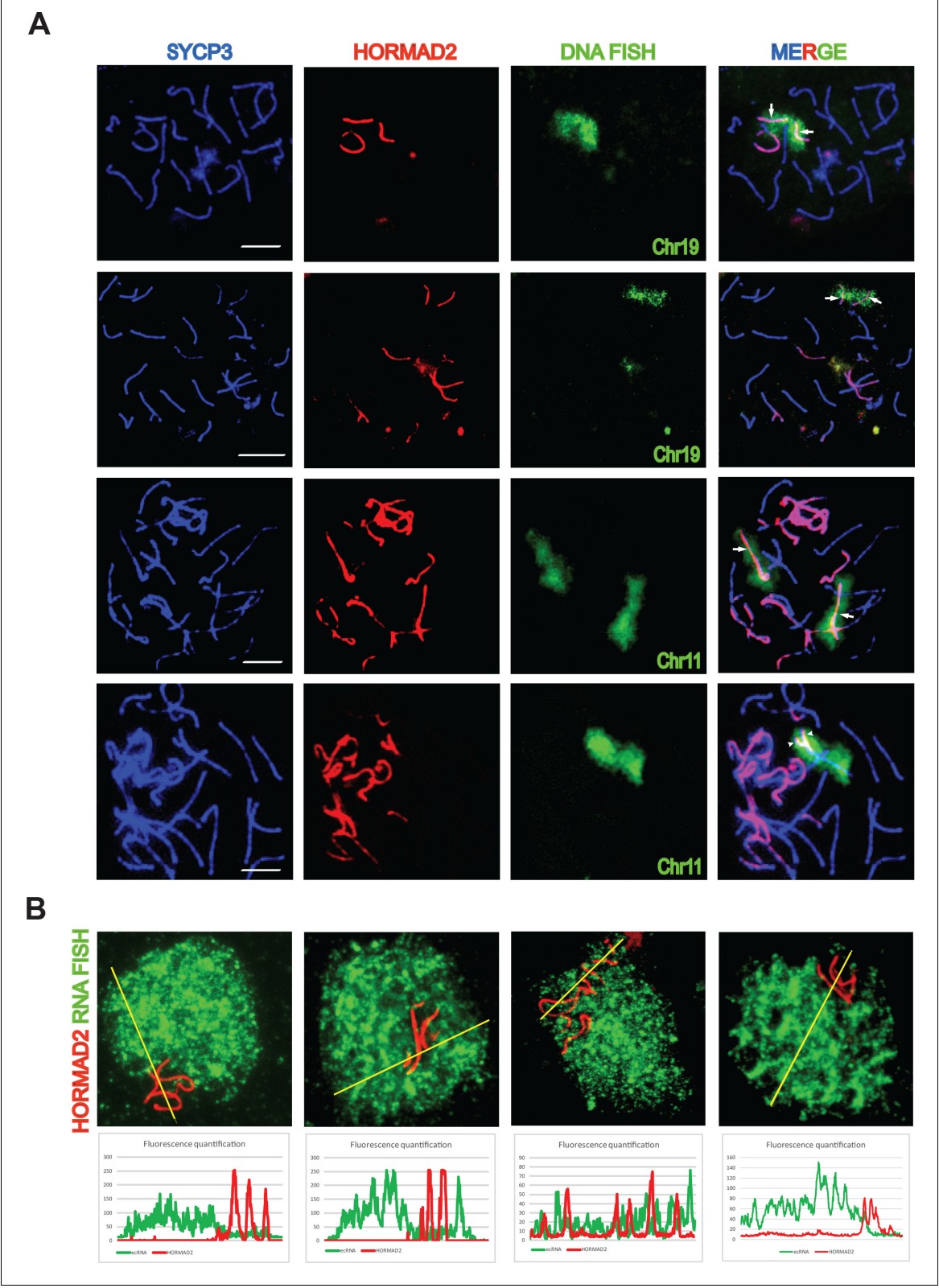

**Figure 1.** Asynapsis of heterosubspecific homologs in PB6F₁ pachynemas. (**A**) Partial (arrowheads) and complete (arrows) asynapsis of Chr 19 and 11. HORMAD2-labeled chromosomes with synapsis defects often form tangles via nonhomologous pairing. Scale bars represent 5 μm. (**B**) Asynapsed chromosomes are embedded in transcriptionally silenced chromatin visualized by the lack of extra-coding RNA (ecRNA) detected by Cot1 RNA FISH. See also *Figure 1—video 1*.

*Figure 1 continued on next page*

*Figure 1 continued*

DOI: https://doi.org/10.7554/eLife.34282.003

The following video is available for figure 1:

**Figure 1–video 1.** The images of the immunofluorescence anti-HORMAD2 stained and cot-1 RNA FISH-labeled spread spermatocytes were examined, and z-stack series were acquired using a confocal microscope (DMI6000CEL – Leica TCS SP8).

DOI: https://doi.org/10.7554/eLife.34282.004

were examined. All male progeny of the 2-chr crosses were fully sterile, with low testis weight and the absence of sperm in the epididymis. The analysis of data from six recombinant chromosomes revealed the common features described below.

Introduction by recombination of 27 Mb or more of a consubspecific (PWD/PWD) interval into a pair of intersubspecific (PWD/B6) homologs effectively suppressed the asynapsis rate below the baseline of 5% in all six studied autosomes (*Figure 3B*). The efficiency of synapsis rescue was gradual with an apparent change point (*Figure 3B*). To describe the pattern in the data, a segmented regression model was used (see 'Materials and methods'). The model based on the data pooled from all 2-chr crosses was selected as the best model with an estimated change point at 27.14 Mb (19.36; 34.91) (95% CI) (see *Figure 3—source data 7*). The slope of the decrease of asynapsis in the region of consubspecific intervals shorter than 27.14 Mb differed for different chromosomes (p=3 × 10⁻¹¹, F-test). For each chromosome, the asynapsis rate decreased from the maximal value

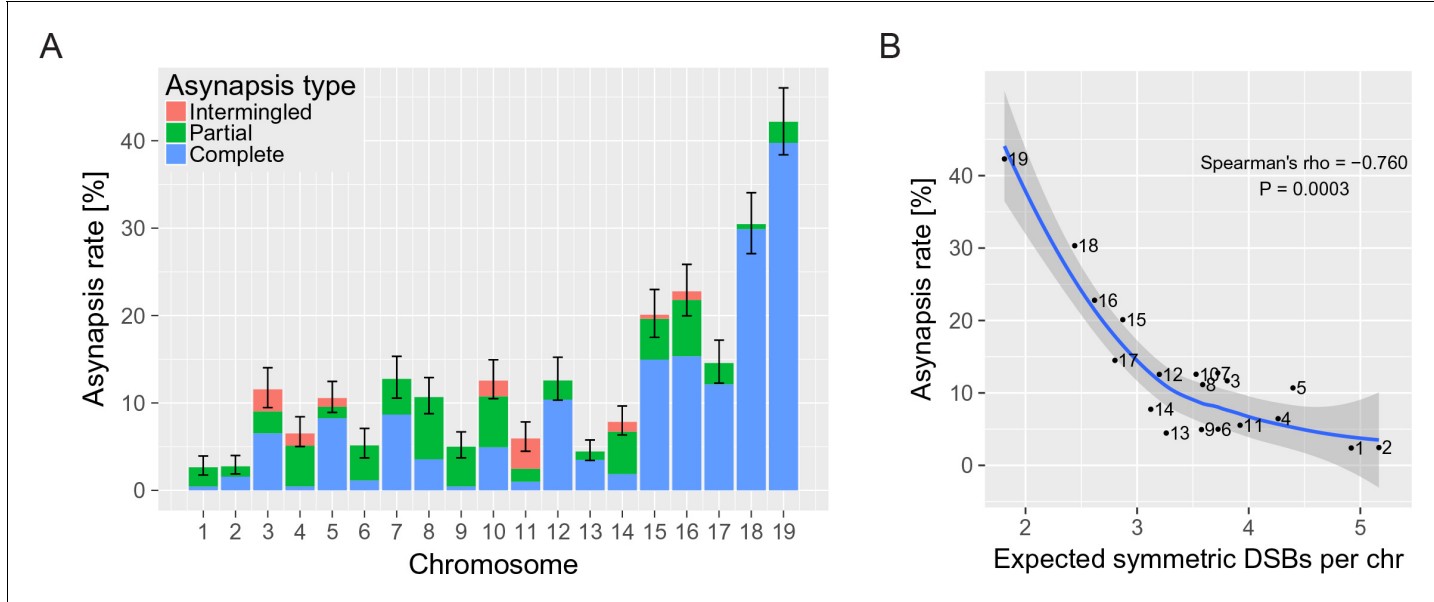

**Figure 2.** The asynapsis rate of individual autosomes in sterile male (PWD x B6)F$_1$ hybrids. (**A**) Mean asynapsis rate ±S.E (based on GLMM model). Intermingled asynapsis refers to overlaps of two or more asynapsed chromosomes within the DNA FISH cloud of chromatin. The five smallest chromosomes had higher asynapsis rate (GLMM model, p=1.1×10⁻¹³). Concurrently, the chromosomes with higher asynapsis rate were also more involved in complete rather than partial asynapsis (GLMM model, p=6.2×10⁻⁵). Proportion of complete and partial asynapsis was controlled by the asynapsis rate rather than by the chromosomal length (test for effect of the length when controlled for the asynapsis rate, p=0.491). (**B**) Negative correlation (Spearman's ρ=−0.760, p=0.0003) between asynapsis rate and mean expected number of symmetric DSBs (*Davies et al., 2016*) based on the chromosome-wide distribution of SPO11 oligos in fertile B6 males (*Lange et al., 2016*).

DOI: https://doi.org/10.7554/eLife.34282.005

The following source data is available for figure 2:

**Source data 1.** Asynapsis rate of individual chromosomes of (PWD x B6)F$_1$ males.

DOI: https://doi.org/10.7554/eLife.34282.006

**Source data 2.** Chromosome-scale comparison of expected DSBs in symmetric hotspots and asynapsis rate.

DOI: https://doi.org/10.7554/eLife.34282.007

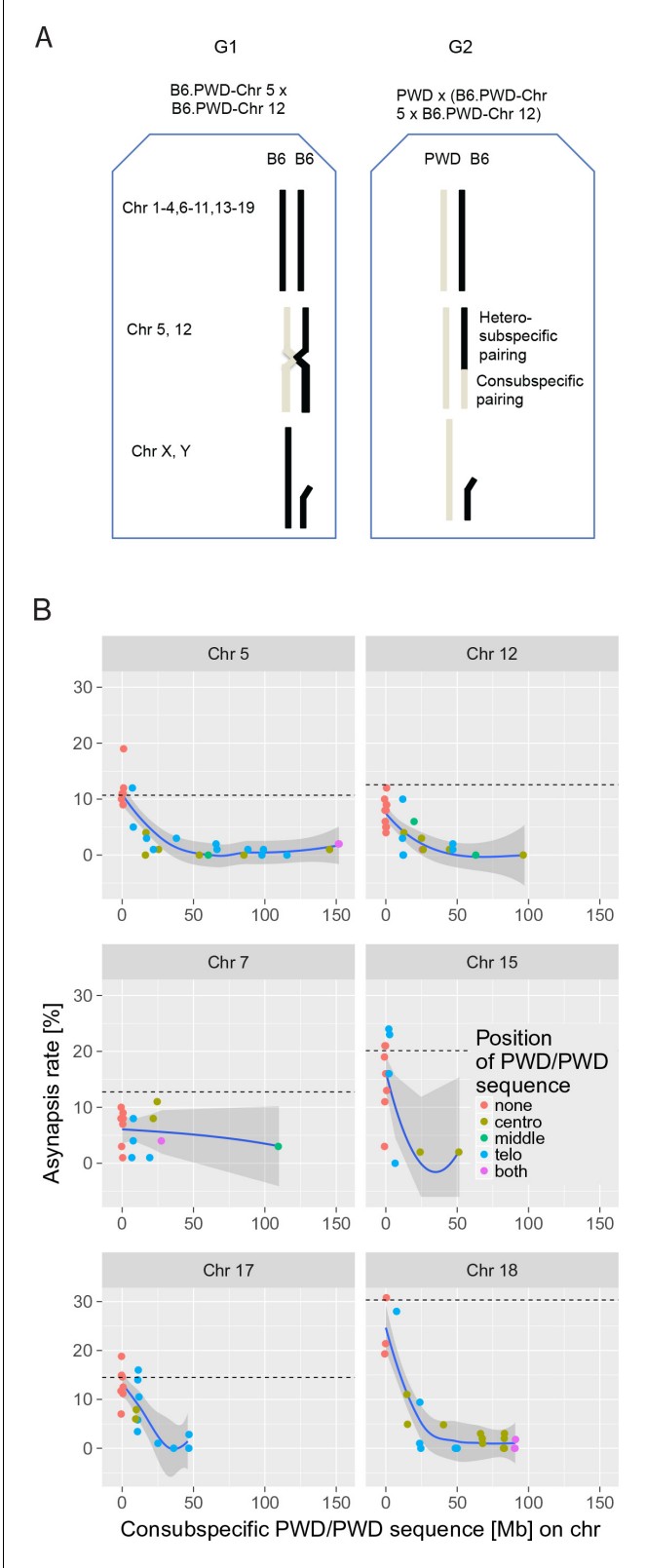

**Figure 3.** The effect of consubspecific PWD/PWD stretches of genomic sequence on pachytene synapsis, 2-chr cross. (A) The F₁ hybrid males of two consomic strains (generation 1 G₁, Chr 5 and Chr 12 shown here) were crossed to PWD females to produce generation 2 G₂ sterile F₁ hybrids with random recombinant consomic chromosomes 5 and 12. Using whole-chromosome probes, the asynapsis rate of the consomic chromosomes was
*Figure 3 continued on next page*

*Figure 3 continued*

scored by DNA FISH. (**B**) Combination of two chromosomes (5 + 12, 7 + 15 and 17 + 18) were challenged in each experiment. The localization of PWD homozygous sequence with respect to centromere, interstitial part of the chromosome or telomere, or on both ends is distinguished by color (see also *Figure 3—source data 1–6*). The average length between the minimum and maximum of the consubspecific sequence is plotted. The mean asynapsis rate of a given chromosome is regularly higher in PB6F$_1$ hybrids (dashed line) than in 2-chr cross. For explanation see *Figure 4* and the section on the *trans*-effect-dependent variation in asynapsis rate. Loess curve with 95% CI.

DOI: https://doi.org/10.7554/eLife.34282.008

The following source data is available for figure 3:

**Source data 1.** The effect of the size and location of PWD/PWD consubspecific intervals on asynapsis of Chr 5.
DOI: https://doi.org/10.7554/eLife.34282.009
**Source data 2.** The effect of the size and location of PWD/PWD consubspecific intervals on asynapsis of Chr 12.
DOI: https://doi.org/10.7554/eLife.34282.010
**Source data 3.** The effect of the size and location of PWD/PWD consubspecific intervals on asynapsis of Chr 7.
DOI: https://doi.org/10.7554/eLife.34282.011
**Source data 4.** The effect of the size and location of PWD/PWD consubspecific intervals on asynapsis of Chr 15.
DOI: https://doi.org/10.7554/eLife.34282.012
**Source data 5.** The effect of the size and location of PWD/PWD consubspecific intervals on asynapsis of Chr 17.
DOI: https://doi.org/10.7554/eLife.34282.013
**Source data 6.** The effect of the size and location of PWD/PWD consubspecific intervals on asynapsis of Chr 18.
DOI: https://doi.org/10.7554/eLife.34282.014
**Source data 7.** Change point estimates of the minimal length [Mb] of PWD/PWD homozygosity showing detectable affect on synapsis rate.
DOI: https://doi.org/10.7554/eLife.34282.015
**Source data 8.** Selected SSLP markers polymorphic between B6 and PWD.
DOI: https://doi.org/10.7554/eLife.34282.016

measured for non-recombinant PWD/B6 (with 0 Mb of PWD/PWD) down below 5% estimated for the change-point value of 27.14 Mb of PWD/PWD interval.

In spite of the known role of subtelomeric (bouquet) association in chromosome pairing (*Ishiguro et al., 2014*; *Scherthan et al., 2014*), the location of the consubspecific sequence at the telomeric end was not essential for synapsis (p=0.9573, F-test). The PWD/PWD intervals of sufficient size rescued synapsis whether located at the centromeric (proximal, n = 14 cases), interstitial (n = 3), or telomeric (distal, n = 14) position (*Figure 3—source data 1–6*).

## Reversal of hybrid sterility by targeted suppression of asynapsis in four of the most asynapsis-sensitive autosomes

The experiments described above have shown that a randomly located consubspecific PWD/PWD interval of 27 or more Mb on otherwise intersubspecific PWD/B6 background is sufficient to restore the pachytene synapsis of a given autosomal pair. To check the causal relationship between meiotic chromosome asynapsis and HS, we attempted to reverse HS by reducing the asynapsis in the four most asynapsis-prone chromosomes. Provided that hybrid male sterility is directly dependent on chromosome synapsis, we predicted (by multiplying the probabilities of the synapsis of individual chromosomes obtained in F$_1$ hybrids) that complete elimination of asynapsis of four of the shortest autosomes (excluding Chr 17 to avoid $Prdm9^{PWD/PWD}$ interference) could increase the proportion of primary spermatocytes that have the full set of synapsed autosomes up to 26.7% and could potentially abolish the apoptosis of these cells to yield around 5 million sperm cells in the epididymis of the hybrid males.

To evaluate this prediction experimentally, random intervals of consomic Chrs 15$^{PWD}$, 16$^{PWD}$, 18$^{PWD}$, and 19$^{PWD}$ were transferred onto the genetic background of B6 mice in a four-generation cross as shown in *Figure 4A*. Eleven G$_3$ males selected for maximal extent of PWD sequence on these chromosomes were crossed to PWD females (*Figure 4—source data 1*). The resulting G$_4$ hybrid male progeny (hereafter referred to as a 4-chr cross) displayed various degrees of PWD homozygosity in the studied consomic autosomes on an otherwise intersubspecific PWD/B6 genetic

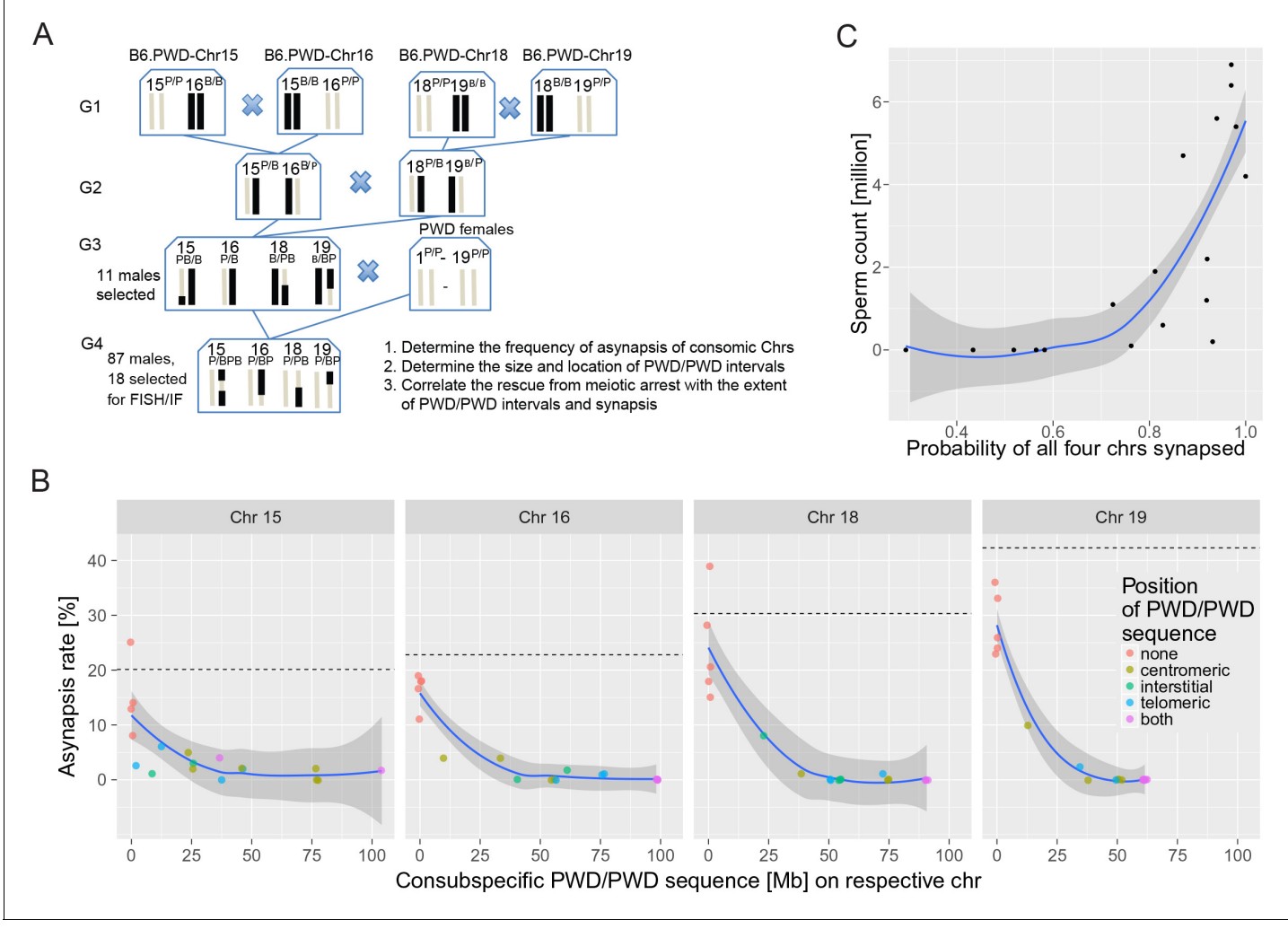

**Figure 4.** The effect of consubspecific PWD/PWD stretches of genomic sequence on pachytene synapsis and meiotic progression, 4-chr cross. (**A**) Scheme of a four-generation cross resulting in F₁ hybrids with four recombinant consomic chromosomes. (**B**) The asynapsis rate related to the size and chromosomal position of the consubspecific PWD/PWD sequence in four consomic chromosomes (15, 16, 18 and 19, see also *Figure 4—source data 3*). The localization of PWD homozygous sequence with respect to the centromere, the interstitial part of the chromosome, or the telomere, or on both ends is distinguished by color (see also *Figure 4—source data 3*). (**C**) Number of sperm in epididymis is a function of the probability of synapsis of all four consomic chromosomes. The complete meiotic arrest is reversed in males having 70% or higher chance of all four chromosomes synapsed. See *Figure 6—figure supplement 2*. Loess curve with 95% CI.

DOI: https://doi.org/10.7554/eLife.34282.017

The following source data and figure supplement are available for figure 4:

**Source data 1.** Eleven G₃ male parents selected for the 4-Chr cross experiment.
DOI: https://doi.org/10.7554/eLife.34282.019

**Source data 2.** The fertility parameters of hybrids of the 4-chr cross experiment.
DOI: https://doi.org/10.7554/eLife.34282.020

**Source data 3.** Four-chr cross.
DOI: https://doi.org/10.7554/eLife.34282.021

**Figure supplement 1.** Fertility parameters of G₄ males from the 4-chr cross.
DOI: https://doi.org/10.7554/eLife.34282.018

background. As predicted, a significant fraction of hybrid males did indeed show partial rescue of spermatogenesis. In the PB6F1 cross, 100% of hybrid males displayed no sperm in the epididymis, whereas in the 4-chr cross, only 51.7% of 87 G₄ males were aspermic, 19.5% had a $0.01–0.74 \times 10^6$

sperm count, and 28.7% had $1.0–13.7 \times 10^6$ sperm cells (*Figure 4—figure supplement 1*, *Figure 4—source data 2*).

Next, we asked whether the reversal of meiotic arrest correlates with the recovery of meiotic synapsis of recombined chromosomes and with the size of PWD/PWD consubspecific stretches in the four manipulated chromosomes. Eighteen $G_4$ males were deliberately selected according to their fertility parameters, 13 with HS partial rescue, displaying sperm cells in the epididymis ($0.1 \times 10–6.9 \times 10^6$), and five aspermic controls. The meiotic analysis of over 6500 pachynemas from the genotyped males confirmed the prediction based on the results of 2-chr crosses. The nonrecombinant PWD/PWD consubspecific bivalents were always fully synapsed, whereas all nonrecombinant PWD/B6 intersubspecific pairs revealed the highest frequencies of asynapsis. All recombinant chromosomes with consubspecific intervals of sufficient length (*Figure 4—source data 3*; see *Figure 3—source data 7* for change point estimates) effectively restored synapsis. Moreover, the presence of sperm cells corresponded with the rescue of synapsis of consomic chromosomes. As a rule of thumb, the hybrids had sperm when asynapsis was suppressed in at least three of four segregating chromosomes and when the probability of all four consomic chromosomes being synapsed was >0.7 (p=0.0014, Mann-Whitney test). Chrs 16, 18, and 19 contributed the strongest effect (*Figure 4—source data 3*).

## Evidence for a *trans* effect on the rate of asynapsis

Provided that the probability of failure of the synapsis of each chromosome was completely independent of the rest of the hybrid genome, then the asynapsis rate of a particular nonrecombinant intersubspecific chromosome pair would be the same in $F_1$ hybrids, 2-chr crosses, and the 4-chr cross. Moreover, the frequency of pachynemas with all chromosomes synapsed could be predicted by multiplication of the observed frequencies of the synapsis of individual chromosomes (see *Figure 2—source data 2*). Such predicted values would be close to the values directly read from the meiotic spreads and would lie along the diagonal in *Figure 5*. As shown below, both types of analysis clearly revealed that the asynapsis rate of a particular chromosome depends on the synapsis status of other chromosomes. First, in PB6F$_1$ hybrids, the observed 13.1% (11.4–14.9%) (95% CI) of fully synapsed pachynemas was double (p=0.023, Mann-Whitney test) the 6.6% (5.3–8.1%) rate expected by the multiplication of the observed synapsis rates of individual chromosomes (*Figure 5—source data 1*), indicating a *trans* effect of synapsed autosomes on the probability of the asynapsis of other PWD/B6 chromosome pairs. The *trans* effect was more pronounced in 2-chr cross and 4-chr cross experiments. Second, at the level of individual chromosomes, the most straightforward comparison was between the nonrecombinant PWD/B6 chromosomes, where the asynapsis rate was dramatically reduced in 2-chr crosses or the 4-chr cross (odds ratio [OR]=0.687, p=0.0002, GLMM) compared to $F_1$ hybrid rates. The *trans* effect was analyzed further for Chromosomes 15, 16, 18 and 19 by comparing the asynapsis rate of a given non-recombinant PWD/B6 pair with the other three analyzed chromosomes in the 4-chr cross and in $F_1$ hybrids. The Supplement 1 to *Figure 5* shows a negative correlation from r=−0.45 for Chr 16 to r =−0.88 for Chr 15. On average, if the predicted synapsis rate of three chromosomes is increased by ten percent, we can expect a 4.18% 2.72–5.34%) (95% CI) decrease of asynapsis rate of the fourth chromosome (p=0.0266, log-log regression). However, for the chromosomes with at least 34.9 Mb of PWD/PWD segment (right bound of 95% CI of change point estimate), for which an additional length of PWD/PWD was not shown to affect asynapsis rate anymore, the *trans* effect could not be detected (p=0.186, comparison of GLMM models).

To conclude, the *trans* effect is the second non-genic effect modifying the asynapsis rate primarily caused by the *cis*-acting inter-homolog incompatibility in PB6F$_1$ primary spermatocytes. The significance and the magnitude of the *trans* effect depends on the *cis*-acting inter-homolog incompatibility.

## Discussion

The genic control of HS and meiotic synapsis in PB6F$_1$ hybrids can be demonstrated by complete restitution of fertility and meiotic pairing in males with *Prdm9*$^{PWD/PWD}$ or *Prdm9*$^{PWD/B6Hu}$ genotypes and by partial recovery in *Prdm9*$^{PWD/C3H}$ males (*Dzur-Gejdosova et al., 2012*; *Bhattacharyya et al., 2013*; *Davies et al., 2016*), but a chromosome-autonomous nature of asynapsis became apparent in experiments where PB6F$_1$ hybrids carried a single pair of PWD/PWD consubspecific homologs. The

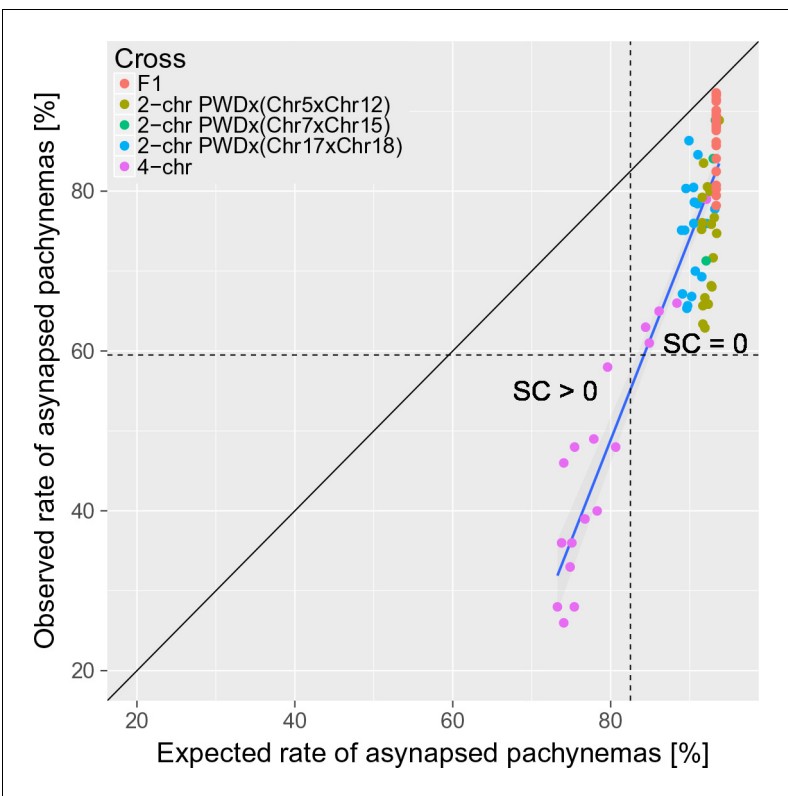

**Figure 5.** The trans-acting effect of consubspecific PWD/PWD stretches increases the probability of full synapsis of PWD/B6 intersubspecific homologs in males of 2-chr cross and 4-chr cross experiments. The expected rate of synapsed pachynemas was calculated for each mouse in 2-chr cross and 4-chr cross experiments by multiplication of observed synapsis rates (i.e. assuming independence) of FISH analyzed chromosomes (e.g. Chrs 15, 16, 18 and 19 in 4-chr cross) and the observed PB6F$_1$ synapsis rates of the remaining autosomes. Asynapsis rate was calculated as a complement to synapsis rate. The difference between expected and observed overall asynapsis is most pronounced in 4-chr cross males with the lowest expected overall asynapsis rate. Recovery of spermatogenesis signaled by the presence of sperm in the epididymis occurs when more than 40% of pachynemas are fully synapsed. SC is sperm count.

DOI: https://doi.org/10.7554/eLife.34282.022

The following source data and figure supplement are available for figure 5:

**Source data 1.** Four-chr cross experiment.

DOI: https://doi.org/10.7554/eLife.34282.024

**Figure supplement 1.** Asynapsis rate of individual nonrecombinant consomic PWD/B6 chromosomes modified in trans by the probability of synapsis of the remaining three consomic chromosomes in the 4-chr cross in individual males compared to PB6F$_1$ hybrids.

DOI: https://doi.org/10.7554/eLife.34282.023

males remained sterile, but the synapsis of the particular consubspecific pair was completely restored (*Bhattacharyya et al., 2013*). Such regulation of meiotic asynapsis in PB6F$_1$ hybrids can be explained by a combined effect of the chromosome-autonomous interaction of homologs operating in *cis* and *Prdm9/Hstx2* incompatibility operating in *trans*. Here, we separated the non-genic chromosome autonomous from genic control mechanisms by keeping the sterility-determining allelic combination of the *Prdm9*$^{PWD}$/*Prdm9*$^{B6}$ gene and *Hstx2*$^{PWD}$ locus constant in all crosses, while successively introgressing stretches of the PWD/PWD consubspecific sequence into eight PWD/B6 intersubspecific autosomal pairs.

## The meiotic asynapsis rate correlates with the presumed paucity of symmetric DSB hotspots in individual chromosomes in sterile hybrids

*Davies et al. (2016)* found that the DNA-binding zinc-finger domain of the PRDM9 molecule is responsible for sterility in $PB6F_1$ hybrids. Further, they found that in the sterile hybrids, most $PRDM9^{PWD}$-specific hotspots reside on B6 chromosomes and, vice versa, that most of the $PRDM9^{B6}$-binding sites are activated on PWD chromosomes. This asymmetry could be explained in part by erosion of the PRDM9-binding sites due to preferential transmission to progeny of the altered hotspots motifs (*Boulton et al., 1997*; *Myers et al., 2010*). In a parallel study, *Smagulova et al. (2016)* identified a novel class of strong hotspots in $PB6F_1$ hybrids that are absent in PWD and B6 parents and that are apparently related to asymmetric hotspots described by *Davies et al. (2016)*. Moreover, *Prdm9*-independent 'default' hotspots were particularly enriched in Chr X, and we noticed that the percentage of these 'default' hotspots in autosomes correlates with the present data on asynapsis rate in $F_1$ hybrids (Spearman's ρ=0.69, p=0.0012). These *Prdm9*-independent hotspots may represent the late-forming DSBs on unsynapsed chromosomes and, as such, they may be a consequence rather than the cause of meiotic asynapsis (see *Kauppi et al. [2013]*).

We found that meiotic asynapsis affects each autosomal pair in $PB6F_1$ intersubspecific hybrids at distinctively unequal rates, with shorter chromosomes affected more often than longer ones. A similar pattern of higher sensitivity of smaller autosomes to the synapsis failure was observed in mice with lowered dosages of SPO11 (*Kauppi et al., 2013*) and in the consequent two-fold DSB reduction. The fact that the asynapsis rate of sterile $F_1$ hybrids correlates better with SPO11-oligo-derived DSB density (inferred from B6 mouse strain data *[Lange et al., 2016]*) than with the chromosome length bringsexperimental support for the idea (*Davies et al., 2016*) linking the asynapsis in sterile $PB6F_1$ hybrids to an insufficient number of symmetric DSB hotspots.

## Small stretches of consubspecific sequence restore the synapsis of intersubspecific chromosomes

Provided that a shortage of symmetric hotspots (*Davies et al., 2016*) is the ultimate cause of the failure of meiotic synapsis of intersubspecific homologs, then the full synapsis could be restored by exchanging the asymmetric hotspots for the symmetric ones. To test this prediction experimentally, we constructed pairs of PWD/B6 intersubspecific homologs carrying stretches of PWD/PWD consubspecific intervals, which by definition cannot carry asymmetric hotspots. We found that chromosomes with 27 Mb or longer stretches of consubspecific sequence always rescued full synapsis in hybrid males. The position of the consubspecific interval along the chromosome was not critical for synapsis rescue, in accordance with the finding that synaptonemal complexes nucleate at multiple recombination sites in each chromosome (*Zickler and Kleckner, 2015*; *Finsterbusch et al., 2016*). We assume that the presence of symmetric DSB in the PWD/PWD homozygous stretches exceeded the threshold of a minimum number of timely repaired DSBs, thus rescuing normal meiotic synapsis.

Allowing for the assumptions enumerated in the 'Results' section, the number of DSBs necessary for proper synapsis of a given chromosome can be estimated on the basis of the expected distribution of symmetric DSB hotspots on all autosomes and their asynapsis ratios in sterile $F_1$ hybrids (*Figure 2—source data 2*). We aimed to model how the induction and repair of DSBs influence proper meiotic synapsis, and tried to estimate the minimum number of symmetric DSBs per chromosome sufficient for full meiotic synapsis. Our model predicts that in approximately 25% of cases, a chromosome is asynapsed because there are only asymmetric DSBs and no symmetric DSBs (slope of the regression of P[asynapsis] on P[0 symmetric DSBs]=4.22). Assuming a critical threshold of the required DSBs, the remaining 75% of asynapsis could occur on chromosomes with one symmetric DSB and with the other DSBs being asymmetric. Thus, it is consistent with our data that two symmetric DSBs per chromosome could be sufficient for full development of the synaptonemal complex, as shown in *Figure 6* (slope of the regression of P[asynapsis] on P[0 or one symmetric DSBs]=1.00). The same conclusion also holds true for 2-chr cross and 4-chr cross experiments (*Figure 6—figure supplements 1* and *2*). The deviations from the diagonal in *Figure 6* depicting the 4-chr cross can be explained by the *trans*-effect described in the 'Results' section.

The *trans* effect as reported in this paper refers to the enhanced probability of synapsis of a pair of intersubspecific homologs depending on successful pairing of other chromosomes in males with the $Prdm9^{PWD}/Prdm9^{B6}$, $Hstx2^{PWD}$ 'hybrid sterility' genotype. The mechanism of the *trans* effect is

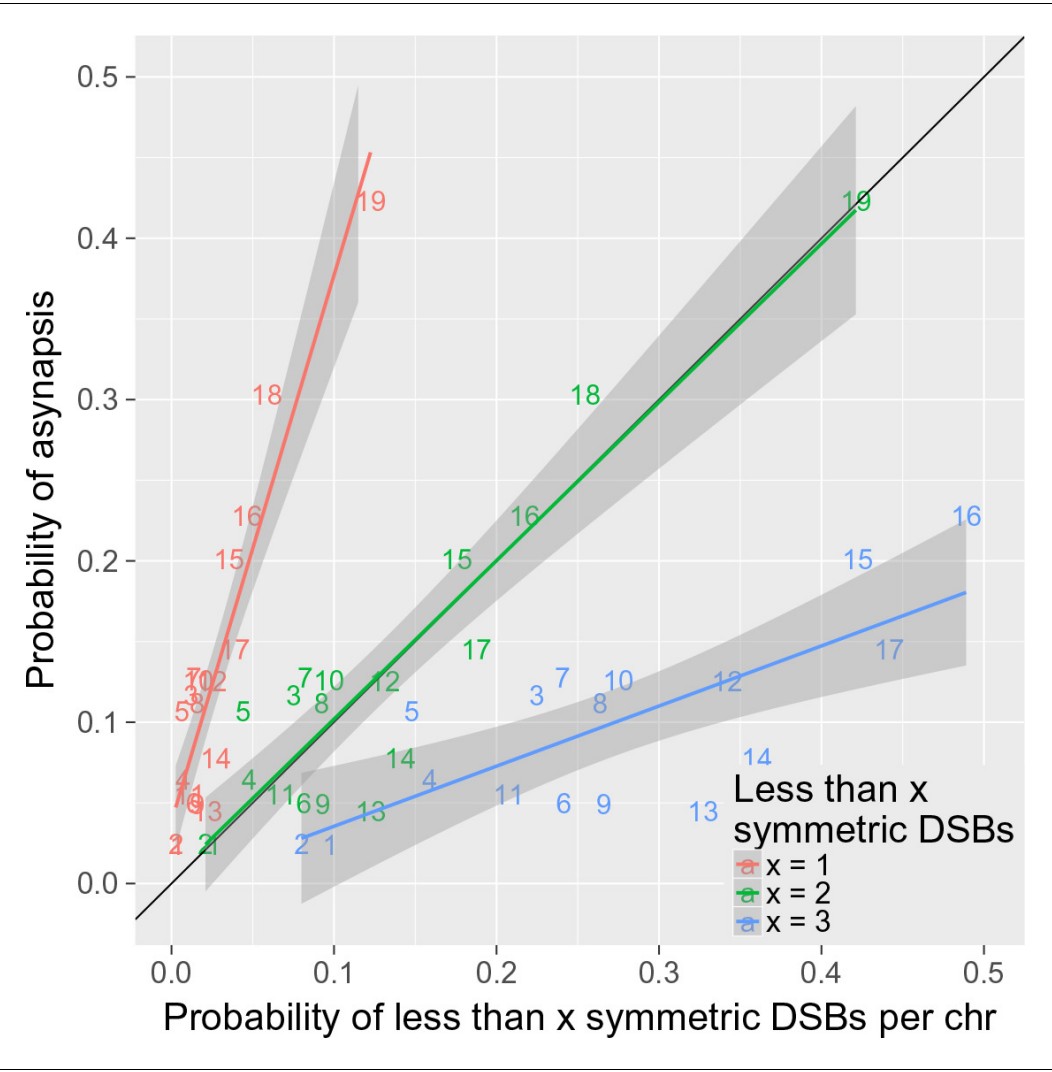

**Figure 6.** Two or more symmetric DSBs can be sufficient for synapsis. The probability of less than one symmetric DSB per chromosome is ~4 times lower than the asynapsis rate observed in PB6F$_1$ hybrid males (i.e. the estimate of probability of asynapsis), implying that ~75% of all asynapsis occurs when there is one or more repairable DSBs. The probability of less than two symmetric DSBs is a good estimate of the probability of asynapsis, whereas the probability of less than three symmetric DSBs overestimates the probability of asynapsis. This shows that, in the simplest explanation, two or more symmetric DSBs could be sufficient for synapsis. The probabilistic distribution of symmetric DSBs is calculated on the basis of the model described in the 'Discussion'.

DOI: https://doi.org/10.7554/eLife.34282.025

The following figure supplements are available for figure 6:

**Figure supplement 1.** Two and more DSBs within symmetric hotspots can be sufficient for synapsis in 2-chr cross males.

DOI: https://doi.org/10.7554/eLife.34282.026

**Figure supplement 2.** Two or more DSBs within symmetric hotspots can be sufficient for synapsis in 4-chr cross hybrid males.

DOI: https://doi.org/10.7554/eLife.34282.027

unknown. *Kauppi et al. (2013)* discussed the chain reactions of asynapsis, in which asynapsis of one or more chromosomes observed in male mice with lowered dosage of SPO11 increased the risk of asynapsis of other chromosomes by engaging them in nonhomologous synapsis among themselves or with the non-PAR region of X chromosome. An alternative explanation of the *trans* effect could involve an unspecified rate- or time-limiting step involving the DSB repair machinery.

The engagement of the X chromosome in autosomal asynapsis could also be related to the fertility of female hybrids (*Forejt, 1996*; *Kauppi et al., 2013*). PB6F$_1$ hybrid females are fertile, obeying Haldane's rule (*Haldane, 1922*) but their oogenesis is impaired, having 45% of pachynemas with one or more asynapsed autosomes. Nevertheless, the effect of *Prdm9* on asynapsis in female PB6F$_1$ meiosis seems weak or absent (*Bhattacharyya et al., 2013, 2014*).

## On the chromosomal nature of hybrid sterility

The vast majority of literature on the genetic mechanism of infertility of inter(sub)specific hybrids focuses on the genetic mapping of hybrid sterility genes, their possible epistatic incompatibilities and evolutionarily diverged structure or expression pattern (*Maheshwari and Barbash, 2011*; *Presgraves, 2010*; *Civetta, 2016*; *Mack and Nachman, 2017*). Likewise, our early studies considered univalents in PB6F1 primary spermatocytes as a secondary consequence of meiotic arrest caused by genic incompatibilities (*Forejt and Iványi, 1974*; *Forejt, 1996*). However, quantitative meiotic analyses revealed that 90% of primary spermatocytes carry one or more pairs of asynapsed homologs and, more importantly, that asynapsis is chromosome-autonomous, depending on inter-homolog (*cis-*) interactions (*Bhattacharyya et al., 2013, 2014*). The findings described in this paper provide the first direct link between *Prdm9*-controlled asynapsis and meiotic arrest in PB6F$_1$ male hybrids. We showed that by deliberately manipulating the synapsis of homologous chromosomes, we could modify the extent of meiotic arrest in intersubspecific PB6F$_1$ hybrids in a predictable way.

Admittedly, the exact molecular basis of meiotic asynapsis and subsequent spermatogenic arrest in PB6F$_1$ males is still unclear and the lack of symmetric DSBs is not necessarily the only explanation for the sterility of thePB6F$_1$hybrid. For instance, multimerization of PRDM9 mediated by PRDM9's zinc fingers (*Baker et al., 2015*; *Altemose et al., 2017*) could alter its DNA-binding properties and enable the default PRDM9-independent hotspots (*Smagulova et al., 2016*) to appear and to generate the *Prdm9* null-like phenotype (*Hayashi et al., 2005*). Recent identification of DSB hotspots within repetitive sequences (*Yamada et al., 2017*) indicate another potential threat to homologous synapsis in intersubspecific hybrids, which could be caused by illegitimate interactions with nonhomologous chromosomes or by the absence of an allelic PRDM9 binding site in the genome of the other subspecies.

Models of the molecular mechanism of hybrid sterility will need to take into consideration that all intersubspecific pairs of homologs synapse properly in the majority of pachynemas. A particular chromosome fails to synapse in as few as 2% and maximally in 45% of PB6F1 pachynemas. Oddly, the B6 allele of the X-linked *Hstx2* locus dramatically increases the pairing efficiency but causes only a small change, if any alteration, in the profile of asymmetric hotspots (*Davies et al., 2016*; *Smagulova et al., 2016*).

The asymmetric DSBs could affect meiotic pairing by hindering repair when searching for the allelic site of a homologous chromosome as a template. The homologous sequence may be inaccessible because of its inappropriate chromatin conformation, such as lack of trimethylation of Lysine four and Lysine 36 of histone H3 or because a critical alteration of the PRDM9-binding motif may provoke the antirecombination activity of the mismatch repair machinery (*Chakraborty and Alani, 2016*) to prevent the repair. It is also probable that some 'difficult' DSBs could be repaired using sister chromatid as a DNA template during the delayed phase of repair when the nonhomologous compensatory synapsis can occur and when the non PAR X chromosome DSBs are most probably repaired (*Kauppi et al., 2013*). However, such inter-sister recombination cannot contribute to the homolog's synapsis.

Hybrid sterility, as well as pairing of homologous chromosomes and meiotic recombination, are universal biological phenomena common to the majority of sexually reproducing organisms. We hypothesize that meiotic pairing and hybrid sterility controlled by *Prdm9* could represent a special case of a more universal reproductive isolation mechanism that is based on meiotic recombination. It is tempting to speculate that the mechanisms that safeguard recombination between homologous allelic sequences could function as checkpoints that disable recombination after homologous sequences have diverged sufficiently during the isolation of closely related taxa. Originally, such an inter-species barrier was proposed by Radman and colleagues (*Rayssiguier et al., 1989*; *Stambuk and Radman, 1998*) to prevent homeologous recombination between *Escherichia coli* and *Salmonella typhiimurium*. Among eukaryotes, the role of the mismatch repair system in reproductive isolation has been reported in *Saccharomyces* species (*Hunter et al., 1996*; *Greig et al., 2003*; *Liti et al., 2006*). An exciting

possibility, which is experimentally testable, posits an antirecombination machinery as a means to gradually restrict gene flow between related taxa, a 'cause in fact' of speciation.

## Materials and methods

### Key resources table

| Reagent type (species) or resource | Designation | Source or reference | Identifiers | Additional information |
|---|---|---|---|---|
| Antibody - primary | Anti SYCP3 (mouse monoclonal, clone D-1) | Santa Cruz Biotechnology | sc-74569; SCP-3 Antibody (D-1); RRID:AB_2197353 | (1:50) |
| Antibody - primary | Anti gH2AFX (rabbit polyclonal) | Abcam | ab2893; gH2AFX antibody; RRID:AB_303388 | (1:1000) |
| Antibody - primary | Anti HORMAD2 (rabbit polyclonal) | DOI: 10.1371/journal.pgen.1000702 | Gift from Dr. Attila Toth | (1:700) |
| Antibody - primary | Anti HORMAD2 (rabbit polyclonal) | Santa Cruz Biotechnology | sc-82192; HORMAD2 antibody (C-18); RRID:AB_2121124 | (1:500) |
| Antibody - secondary | Anti-Rabbit IgG - AlexaFluor568 (goat polyclonal) | Molecular Probes | A-11036; RRID:AB_10563566 | (1:500) |
| Antibody - secondary | Anti-Mouse IgG - AlexaFluor647 (goat polyclonal) | Molecular Probes | A-21235; RRID:AB_2535804 | (1:500) |
| Blocking reagent for immunostaining | Normal goat serum from healthy animals | Chemicon | S26-100ML | |
| Protease inhibitors | Complete, Mini, EDTA-free Protease Inhibitor Cocktail | Roche | 4693159001 | |
| Paraformaldehyde | Paraformaldehyde AQ solution | Electron Microscopy Sciences | 15714S | |
| DNA-FISH probes for mouse chromosomes 1–19 | XMP X Green - Mouse chromosome paints | MetaSystems | D-1401–050-FI; D-1420–050-FI | |
| RNA FISH | Mouse Cot-1 DNA | Invitrogen | 18440016 | |
| RNA FISH | Biotin nick translation kit | Roche | 11,745,824,910 | |
| RNA FISH | Biotinylated goat anti-avidin antibody | Vector Laboratories | BA-0300; RRID:AB_2336108 | (1:100) |
| RNA FISH | Fluorescein-Avidin-DCS | Vector Laboratories | A-2011; RRID:AB_2336456 | (1:100) |
| RNase inhibitor | Ribonucleoside-vanadyl complex (RVC) | SIGMA - ALDRICH | 94742 | (1:100 = 2 mM) |

### Mice, ethics statement and genotyping

The mice were maintained at the Institute of Molecular Genetics in Prague and Vestec, Czech Republic. The project was approved by the Animal Care and Use Committee of the Institute of Molecular Genetics AS CR, protocol No 141/2012. The principles of laboratory animal care, Czech Act No. 246/1992 Sb., compatible with EU Council Directive 86/609/EEC and Appendix of the Council of Europe Convention ETS, were observed. Simple sequence length polymorphisms (SSLP) markers used for genotyping consomic chromosomes in 2-chr crosses and 4-chr cross are listed in *Figure 3—source data 8*. The PWD/Ph inbred strain originated from a single pair of wild mice of the *Mus musculus musculus* subspecies trapped in 1972 in Central Bohemia, Czech Republic (*Gregorová and Forejt, 2000*). The C57BL/6J (B6) inbred strain was purchased from The Jackson Laboratory. The panel of 27 chromosome substitution strains C57BL/6J-Chr #PWD (abbreviated B6.PWD-Chr #) was prepared in our laboratory (*Gregorová et al., 2008*) and is maintained by the Institute of Molecular Genetics AS CR, Division BIOCEV, Vestec, Czech Republic, and by The Jackson Laboratory, Bar Harbor, Maine, USA. All mice were maintained in a 12 hr light/12 hr dark cycle in a specific pathogen-free barrier facility. The mice had unlimited access to a standard rodent diet (ST-1, 3.4% fat; VELAZ) and water. All males were killed at age 60–70 d.

## Immunostaining and image capture

For immunocytochemistry, the spread nuclei were prepared as described (*Anderson et al., 1999*) with modifications. Briefly, a single-cell suspension of spermatogenic cells in 0.1M sucrose with protease inhibitors (Roche) was dropped on 1% paraformaldehyde-treated slides and allowed to settle for 3 hr in a humidified box at 4°C. After brief $H_2O$ and PBS washing and blocking with 5% goat sera in PBS (vol/vol), the cells were immunolabeled using a standard protocol with the following antibodies: anti-HORMAD2 (1:700, rabbit polyclonal antibody, a gift from Attila Toth) and SYCP3 (1:50, mouse monoclonal antibody, Santa Cruz, #74569). Secondary antibodies were used at 1:500 dilutions and incubated at room temperature for 60 min: goat anti-Rabbit IgG-AlexaFluor568 (MolecularProbes, A-11036) and goat anti-Mouse IgG-AlexaFluor647 (MolecularProbes, A-21235). The images were acquired and examined using a Nikon Eclipse 400 microscope with a motorized stage control using a Plan Fluor objective, 60x (MRH00601; Nikon) and captured using a DS-QiMc monochrome CCD camera (Nikon) and the NIS-Elements program (Nikon). The images were processed using the Adobe Photoshop CS software (Adobe Systems).

## Combined immunofluorescence staining with DNA FISH or RNA FISH

XMP XCyting Mouse Chromosome N Whole Painting Probes (Metasystems) were used for the DNA FISH analysis of asynapsis of all autosomes, one at a time, as described (*Turner et al., 2005*), with slight modifications. Testes from 8-week-old mice were dissected and spread meiocyte nuclei were prepared as described previously (*Mahadevaiah et al., 2009*) with a modification, which relies on a reversed sequence of RNA FISH and immunofluorescence staining. Briefly, after cell fixation and permeabilization, the immunofluorescent labeling was performed for 90 min at 20°C with primary anti-HORMAD2 and anti-SYCP3 antibodies. Secondary antibodies were selected as above and incubated at room temperature for 60 min. After washing and postfixation steps, the immunostained nuclei were processed with RNA fluorescence in situ hybridization. The Cot-1 DNA biotin-labeled probe was incubated overnight at 37°C, and then the hybridized biotinylated Cot-1 probe was labelled with a FITC–avidin conjugate and the fluorescent signal was amplified as described previously (*Chaumeil et al., 2008*). The images of the immunofluorescence stained and Cot-1 RNA FISH-labeled spread spermatocytes were examined and photographed using confocal microscope DMI6000CEL – Leica TCS SP8.

## Statistics

To model the dependence between the asynapsis rate and the number of symmetric DSBs, we determined the probabilistic distribution of the number of symmetric DSBs. The distribution was determined by simulation and with parameters based on previous studies. (i) The number of DSBs per cell (*Bhattacharyya et al., 2013*) was modeled as an observation from the normal distribution N (250, sd = 20). (ii) We assumed a number of DSBs proportional to SPO11 oligos (*Lange et al., 2016*) in each autosome. (iii) The positions of DSBs in the particular autosome were simulated from the uniform distribution, U (0, Autosome_length). (iv) For the intersubspecific part of the autosomal pair, the number of symmetric DSBs was simulated from the binomial distribution Bi (n = N_DSBs_in_het_part, p=0.28). For the consubspecific part of the autosome, all DSBs were taken to be symmetric. The total number of symmetric DSBs in the autosome was taken as the sum of symmetric DSBs in the respective parts. Steps (i) to (iv) were performed in N = 100000 simulations to obtain a probabilistic distribution (Source Code 1).

The effects of the number of Spo11 oligos and the chromosomal length on the asynapsis rate were investigated using a GLMM model. In all of the GLMM models used in this work, the asynapsis was modeled as a binary response to the fixed effects under investigation and a random intercept for each animal. In $F_1$ hybrids, 95% confidence intervals for the the observed rates of asynapsed pachynemas and for the expected rated of asynapsed pachynemas were calculated by bootstrap. The estimates of mean asynapsis rate in respective chromosomes, their standard errors and their 95% confidence intervals were based on the GLMM model. In 2-chr crosses and the 4-chr cross, 95% confidence intervals of asynapsis rate were constructed on the basis of the likelihood ratio to also capture the uncertainty in the cases when the zero asynapsis rate per mouse and per chromosome was observed.

On the basis of the nature of the dependence between the asynapsis rate and the length of the consubspecific PWD/PWD region on chromosomes of 2-chr crosses and the 4-chr cross, we fitted the data to segmented two-part continuous regression models (*Muggeo, 2003*).

We fitted the models for all the chromosomes separately (see *Figure 3—source data 7*), being aware of the limitations caused by the lack of animals having specific lengths of the consubspecific PWD/PWD region in the respective chromosomes. As the best model describing the dependence of asynapsis rate on the lengths of PWD/PWD intervals, we selected piecewise linear models fitting 1) pooled data from 2-chr crosses and 2) pooled data from both 2-chr crosses and the 4-chr cross. These models are not severely affected by the lack of observations with specific lengths of the PWD/PWD segment neither by outliers.

All calculations were performed in R 3.2.2 (RRID:SCR_001905); the change point models and the GLMM models were fitted using the packages segmented and lme4, respectively (*Muggeo and Ferrara, 2008*; *Bates et al., 2015*).

## Acknowledgements

We thank Simon Myers for sharing his unpublished data and critical comments, Attila Tóth for providing the HORMAD2 antibody, Mary Ann Handel and Linda Odenthal-Hesse for critical reading of this manuscript, David Green, Sarka Takacova, and members of the Forejt lab for their helpful comments, M Capek for help with analysis of confocal microscopy data, and J Perlova for her assistance with genotyping. This work was supported by Czech Science Foundation grant 13–08078S and by the LQ1604 project of NSPII from the Ministry of Education, Youth and Sports of the Czech Republic. Barbora Valiskova was partly supported by project GA UK No.17115 from Charles University, Czech Republic. We also acknowledge the Light Microscopy Core Facility, IMG ASCR, Prague, which is supported by MEYS (LM2015062) and by OPPK (CZ.2.16/3.1.00/21547).

## Additional information

### Funding

| Funder | Grant reference number | Author |
|---|---|---|
| Grantová Agentura Ceské Republiky | 13-08078S | Jiri Forejt |
| Charles University Grant Agency of The Czech Republic | 435416 | Vaclav Gergelits Barbora Valiskova |
| Ministry of Education, Youth and Sports of The Czech Republic | LQ1604 project of the NSPII | Jiri Forejt |
| Charles University Grant Agency of The Czech Republic | 17115 | Vaclav Gergelits Barbora Valiskova |

The funders had no role in study design, data collection and interpretation, or the decision to submit the work for publication.

### Author contributions

Sona Gregorova, Resources, Data curation, Formal analysis, Supervision, Methodology, Writing—original draft; Vaclav Gergelits, Conceptualization, Data curation, Formal analysis, Investigation, Methodology, Writing—original draft, Writing—review and editing; Irena Chvatalova, Vladana Fotopulosova, Diana Wiatrowska, Formal analysis, Investigation, Methodology; Tanmoy Bhattacharyya, Data curation, Formal analysis, Investigation, Methodology; Barbora Valiskova, Data curation, Formal analysis, Investigation; Petr Jansa, Conceptualization, Formal analysis, Investigation, Methodology, Writing—original draft; Jiri Forejt, Conceptualization, Supervision, Funding acquisition, Validation, Investigation, Writing—original draft, Project administration, Writing—review and editing

## Author ORCIDs

Vaclav Gergelits (iD) http://orcid.org/0000-0002-5178-8833
Jiri Forejt (iD) http://orcid.org/0000-0002-2793-3623

## Ethics

Animal experimentation: The mice were maintained at the Institute of Molecular Genetics in Prague and Vestec, Czech Republic. The project was approved by the Animal Care and use Committee of the Institute of Molecular Genetics AS CR, protocol No 141/2012. The principles of laboratory animal care Czech Act No. 246/1992 Sb., compatible with EU Council Directive 86/609/EEC and Apendix of the Council of Europe Convention ETS, were observed.

## Decision letter and Author response

Decision letter https://doi.org/10.7554/eLife.34282.032
Author response https://doi.org/10.7554/eLife.34282.033

## Additional files

### Supplementary files

• Source code 1. The code in R contains simulations of assumed probabilistic distributions of numbers of symmetric DSBs per chromosome in $F_1$, 2-chr and 4-chr crosses. The probabilities of occurrence of respective numbers of symmetric DSBs are compared to the observed asynapsis rates in animals from $F_1$, 2-chr and 4-chr crosses. This code enables reproduction of *Figure 6*, *Figure 6—figure supplements 1* and *2*.
DOI: https://doi.org/10.7554/eLife.34282.028

• Source code 2. This file, crosses_F1_2chr_4chr.csv, contains the data for Source Code File 1.
DOI: https://doi.org/10.7554/eLife.34282.029

• Transparent reporting form
DOI: https://doi.org/10.7554/eLife.34282.030

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
