## [Decision Letter]

Thank you for submitting your article "Modulation of *Prdm9*-controlled meiotic chromosome asynapsis overrides hybrid sterility in mice" for consideration by *eLife*. Your article has been favorably evaluated by Patricia Wittkopp (Senior Editor) and three reviewers, one of whom, Scott Keeney (Reviewer #1), is a member of our Board of Reviewing Editors. The following individual involved in review of your submission has agreed to reveal their identity: John Schimenti (Reviewer #2).

The reviewers have discussed the reviews with one another and the Reviewing Editor has drafted this decision to help you prepare a revised submission.

Summary:

Speciation can be driven in part by fertility defects in the hybrid offspring derived from crosses between individuals from diverged populations in many species, but detailed mechanisms underlying this hybrid sterility are not understood. Prior work dating back many years from the senior author's laboratory has documented hybrid sterility in crosses of PWD and B6 mouse strains. This impressive study tests the hypothesis that chromosome synapsis defects caused by heterozygosity across large chromosomal segments is a key contributor to sterility, rather than incompatibility of function of specific interacting gene pairs. The experimental approach is to test the effect of introgressing more or less random segments of homozygosity in the context of otherwise heterozygous PWD x B6 chromosome pairs, and evaluating chromosome synapsis and fertility. Two very important findings emerged: 1) that large stretches of conspecific homology along a chromosome greatly improves synapsis rates, and 2) the improved synapsis on one chromosome can improve interspecific synapsis on other chromosomes. Remarkably, the data strongly suggest a minimum threshold of ~27 Mbp for homozygosity being sufficient to support efficient chromosome synapsis. Moreover, restoring homozygosity for portions of four of the five smallest chromosomes is sufficient to rescue fertility in PWD x B6 hybrids. The results are interpreted in light of prior findings from other labs concerning apparent delays in repair of DSBs made within "asymmetric" hotspots, i.e., hotspots with allelic differences in binding by PRDM9. The molecular bases for the two major findings are speculated to be the consequence of increasing symmetric DSBs in item #1, and *very speculatively* by alteration of the mismatch repair machinery, in some unknown fashion, in item #2. These mechanisms would be nice to know, but are reasonably considered to be outside the scope of this study.

This is an intriguing set of data that is well presented and explained (with a few exceptions noted below). The experiments to define synapsis rates for each chromosome were nothing short of heroic. Similarly, the experiments to measure synapsis in consubspecific hybrids were quite a chore, but the result is very important. The paper will be of interest to a broad audience interested in meiosis, chromosome dynamics, and evolutionary mechanisms. There are a number of issues that need to be addressed to improve clarity of presentation or deal better with discussions of published work, but these can all be dealt with by changes to text. No additional experimentation is requested.

Essential revision:

1) Throughout: more care is needed with wording when discussing DSBs and hotspots. In some cases, text specifies DSBs when hotspots are meant, and in other cases, hotspots are mentioned when DSBs are meant. One example is the Abstract: "symmetric DSBs" is not really what is meant (DSBs are not symmetric per se, but rather the chromatin at hotspots is thought to be symmetric owing to biallelic action of PRDM9). Better to say "DSBs within symmetric hotspots". (Also, symmetry hasn't been defined yet, so this will not be understandable in the Abstract.) Another example is in Results, which describes a "correlation.… of asynapsis rate with symmetric DSB hotspots". What is meant is a correlation of asynapsis rate with (predicted) DSBs within symmetric hotspots. Similar concern applies to the end of the subsection “Small chromosomes are more susceptible to asynapsis in sterile F1 hybrids”: the data indicate that there is a requirement for a minimum number of DSBs within symmetric hotspots, not a minimum number of symmetric hotspots. Other examples can be found in the subsections “The meiotic asynapsis rate correlates with the presumed paucity of symmetric DSBs in individual chromosomes in sterile hybrids” and “Small stretches of consubspecific sequence restore the synapsis of intersubspecific chromosomes”.

2) The second section of Results is expected and not novel. It is by now very well established that asynapsis is associated with MSUC. The findings can be reduced to a couple of sentences within the first section, indicating that MSUC accompanies the observed asynapsis, as expected from numerous prior studies.

3) Subsection “On the chromosomal nature of hybrid sterility”, last paragraph: The Discussion implies that "asymmetric DSBs" (really, DSBs in asymmetric hotspots) do not get repaired because of excessive heterology (heteroduplex rejection). Davies et al. proposed instead that it is asymmetry of PRDM9-dependent chromatin remodeling that causes the repair delay. Moreover, Davies et al. provided fairly strong evidence that DNA sequence heterology is unlikely to be the root cause. Also, there is of high degrees of sequence polymorphism within recombining regions any of a large number of compatible F1 hybrids of inbred strains (e.g., in B6 x A/J). It isn't obvious why only those sequence polymorphisms that affect PRDM9 binding would influence the recombination reaction, while all other sequence polymorphisms nearby do not (i.e., polymorphisms in symmetric hotspots). Thus, the proposed contribution of sequence mismatches is not convincing, and needs to be discussed in a more balanced way. Care should be taken to also make clear that even the DSBs in asymmetric hotspots are repaired, because the data do not indicate that unrepaired DSBs are responsible for spermatocyte death. Delayed repair by interhomolog recombination or repair by intersister recombination could be explicitly mentioned.

4) Can the general magnitude of the chromosome synapsis trans-effect be calculated from the data?

5) Subsection “The minimal length of consubspecific sequence necessary to rescue meiotic chromosome synapsis”, last paragraph –: Can the authors provide statistical support for this statement?

6) End of subsection “Reversal of hybrid sterility by targeted suppression of asynapsis of four of the most asynapsis-sensitive autosomes”: Can the authors provide statistical support for this statement?

7) Please move all of the supplemental text (Materials and methods plus references) into the main text.

8) Please move examples of the primary immuno-FISH data (Figure 1—figure supplement 1A) into the main text.

---

## [Author Response]

Essential revision:1) Throughout: more care is needed with wording when discussing DSBs and hotspots. In some cases, text specifies DSBs when hotspots are meant, and in other cases, hotspots are mentioned when DSBs are meant. One example is the Abstract: "symmetric DSBs" is not really what is meant (DSBs are not symmetric per se, but rather the chromatin at hotspots is thought to be symmetric owing to biallelic action of PRDM9). Better to say "DSBs within symmetric hotspots". (Also, symmetry hasn't been defined yet, so this will not be understandable in the Abstract.) Another example is in Results, which describes a "correlation.… of asynapsis rate with symmetric DSB hotspots". What is meant is a correlation of asynapsis rate with (predicted) DSBs within symmetric hotspots. Similar concern applies to the end of the subsection “Small chromosomes are more susceptible to asynapsis in sterile F1 hybrids”: the data indicate that there is a requirement for a minimum number of DSBs within symmetric hotspots, not a minimum number of symmetric hotspots. Other examples can be found in the subsections “The meiotic asynapsis rate correlates with the presumed paucity of symmetric DSBs in individual chromosomes in sterile hybrids” and “Small stretches of consubspecific sequence restore the synapsis of intersubspecific chromosomes”.

We agree with the criticism and thank for the comment. The text was modified to discriminate between symmetric DSB hotspots and DSBs within symmetric hotspots: (we abbreviate, for the sake of clarity, DSBs within symmetric hotspots to symmetric DSBs) "[…]the number of DSBs within symmetric hotspots (hereafter symmetric DSBs)[…]"

2) The second section of Results is expected and not novel. It is by now very well established that asynapsis is associated with MSUC. The findings can be reduced to a couple of sentences within the first section, indicating that MSUC accompanies the observed asynapsis, as expected from numerous prior studies.

We reduced the text and fused it to the first section of the Results, as recommended.

3) Subsection “On the chromosomal nature of hybrid sterility”, last paragraph: The Discussion implies that "asymmetric DSBs" (really, DSBs in asymmetric hotspots) do not get repaired because of excessive heterology (heteroduplex rejection). Davies et al. proposed instead that it is asymmetry of PRDM9-dependent chromatin remodeling that causes the repair delay. Moreover, Davies et al. provided fairly strong evidence that DNA sequence heterology is unlikely to be the root cause.

We agree that reducing the possible explanations to antirecombination effect of MMR was one-sided and so in this revision we elaborated it in more general and balanced way. Yet we strongly believe that the asymmetry of PRDM9-dependent chromatin remodeling can be most likely explained by delayed/imperfect PRDM9 binding due to degenerated binding motif on self chromosome. The erasure of PRDM9 binding motifs by meiotic drive is the essence of Myers and Donnelly's findings of PB6F1 male sterility. In their paper they say "These findings reveal that subspecies-specific degradation of PRDM9 binding sites by meiotic drive, which steadily increases asymmetric PRDM9 binding, has impacts beyond simply changing hotspot positions, and strongly support a direct involvement in hybrid infertility". Moreover, 'heterology' is considered in Smagulova et al. paper as well (“we propose that sequence divergence generated by hot spot turnover may create an impediment for recombination in hybrids, potentially leading to reduced fertility and, eventually, speciation”).

Also, there is of high degrees of sequence polymorphism within recombining regions any of a large number of compatible F1 hybrids of inbred strains (e.g., in B6 x A/J).

We suppose that it is the difference in degree of erasure of binding motifs between subspecies that is likely to generate the asymmetry. Smagulova et al. compared DMC1 hotspot symmetry of a number of intra- and inter- subspecific crosses and found the asymmetry typical of the latter. The erasure of binding motifs is most apparent when genomes of two different subspecies match each other.

It isn't obvious why only those sequence polymorphisms that affect PRDM9 binding would influence the recombination reaction, while all other sequence polymorphisms nearby do not (i.e., polymorphisms in symmetric hotspots).

To our best knowledge, which may be incomplete, the synapsis of homologous chromosomes depends solely on the repair of DSBs via non-sister HR. We are not aware about any report demonstrating the effect of general polymorphisms outside mouse PRDM9 binding motifs on meiotic synapsis of homologous chromosomes.

Thus, the proposed contribution of sequence mismatches is not convincing, and needs to be discussed in a more balanced way. Care should be taken to also make clear that even the DSBs in asymmetric hotspots are repaired, because the data do not indicate that unrepaired DSBs are responsible for spermatocyte death. Delayed repair by interhomolog recombination or repair by intersister recombination could be explicitly mentioned.

We recognize the need that the topic has to be discussed in a more balanced way. We completely re-wrote the critical part, taking into account the comments mentioned above, starting from the second paragraph of the subsection “On the chromosomal nature of hybrid sterility” to the end of Discussion.

4) Can the general magnitude of the chromosome synapsis trans-effect be calculated from the data?

Thank you for raising this question; to answer we included the following text: – “The *trans* effect was analyzed further for chromosomes 15, 16, 18 and 19 by comparing the asynapsis rate of a given nonrecombinant PWD/B6 pair with the other three analyzed chromosomes in 4-chr cross and in F1 hybrids. […] On average if predicted synapsis of three chromosomes is increased by ten percent, we expect 4.18% , (95% CI = 2.72% , 5.34% ) decrease of asynapsis rate of the fourth chromosome (P = 0.0266, log-log regression).”

The primary aim of the study was not to measure the *trans* effect, because we found it during the analysis of our data, and as such it was not designed for this aspect. Nevertheless, there is enough evidence for its presence, as documented by the comparison between expected and observed overall asynapsis rates, and quantified by differences in asynapsis rates on respective chromosomes in F1, 4-chr and 2-chr crosses. It would be rather speculative to extrapolate these observations further, to more general conclusions on the magnitude of the *trans* effect.

5) Subsection “The minimal length of consubspecific sequence necessary to rescue meiotic chromosome synapsis”, last paragraph: Can the authors provide statistical support for this statement?

Statistical support was provided: “the location of the consubspecific sequence at the telomeric end was not essential for synapsis (P = 0.9573, F-test)”.

6) End of subsection “Reversal of hybrid sterility by targeted suppression of asynapsis of four of the most asynapsis-sensitive autosomes”: Can the authors provide statistical support for this statement?

Statistical support was provided: “the probability of all four consomic chromosomes being synapsed was > 0.7 (P = 0.0014, Mann-Whitney test).”

7) Please move all of the supplemental text (Materials and methods plus references) into the main text.

All supplemental text including references was moved to the main text.

8) Please move examples of the primary immuno-FISH data (Figure 1—figure supplement 1A) into the main text.

Figure 1—figure supplement 1 was moved into the main text as Figure 1.